# Attentive Linguistic Tracking in Diffusion Models for Training-free Text-guided Image Editing

Bingyan Liu*
South China University of Technology
Guangzhou, China
eeliubingyan@mail.scut.edu.cn

Chengyu Wang†
Alibaba Group
Hangzhou, China
chengyu.wcy@alibaba-inc.com

Jun Huang
Alibaba Group
Hangzhou, China
huangjun.hj@alibaba-inc.com

Kui Jia†
The Chinese University of Hong Kong
Shenzhen, China
kuijia@cuhk.edu.cn

## Abstract

Building on recent breakthroughs in diffusion-based text-to-image synthesis (TIS), training-free text-guided image editing (TIE) has emerged as an indispensable aspect of modern image editing practices. This technique involves the modification of features within attention layers to alter objects or their attributes within images during the generation process. Despite its utility, current image editing algorithms face challenges, particularly when editing multiple objects in an image. In this paper, we introduce VICTORIA, a novel approach that augments TIE by incorporating linguistic knowledge into the manipulation of attention maps during image generation. VICTORIA capitalizes on mechanisms within self-attention layers to ensure spatial consistency between source and target images. Further, we design a novel loss function that refines cross-attention maps, ensuring their alignment with linguistic constraints, thereby enhancing the editing precision of multiple target objects. We also present a linguistic mask blending technique that aids in the retention of information in regions not subjected to modification. Experimental results across seven diverse datasets show that VICTORIA achieves significant improvements over state-of-the-art methods. Our work underscores the critical role and effectiveness of linguistic analysis in elevating the performance of TIE, with a specific emphasis on multi-object scenarios. [1]

## CCS Concepts

• **Computing methodologies** → **Image processing**.

*Contribution during internship at Alibaba Group
†Co-corresponding authors.

[1]The code is available at https://github.com/alibaba/EasyNLP/tree/master/diffusion/VICTORIA.

## Keywords

training-free method, multi-object image editing, linguistic knowledge, diffusion models

**ACM Reference Format:**

Bingyan Liu, Chengyu Wang, Jun Huang, and Kui Jia. 2024. Attentive Linguistic Tracking in Diffusion Models for Training-free Text-guided Image Editing. In *Proceedings of the 32nd ACM International Conference on Multimedia (MM '24), October 28-November 1, 2024, Melbourne, VIC, Australia.* ACM, New York, NY, USA, 9 pages. https://doi.org/10.1145/3664647.3680594

## 1 Introduction

Text-to-Image Synthesis (TIS) has emerged as a groundbreaking field at the intersection of computer vision and natural language processing (NLP), offering the capability to generate visually compelling images from textual descriptions. Pioneering models such as Stable Diffusion [28], DALL-E 2 [26], Imagen [30], and the more recent DALL-E 3 [21], have demonstrated an exceptional ability to produce artistically coherent images. This development has attracted significant attention and research interest from both the academic community and industry [21, 35].

Training-free text-guided image editing (TIE) has emerged as a significant research area, avoiding the necessity for extensive training on large datasets. Instead, this approach harnesses pre-trained Text-to-Image Synthesis (TIS) models or algorithms to edit images directly using textual prompts. Current training-free TIE techniques [1, 2, 5, 10, 16–18, 23, 33] excel in tasks such as image translation, style transformation, and modification of visual attributes, all the while preserving the structural and compositional integrity of the source images. Notably, Prompt-to-Prompt (P2P) [10] skillfully modifies specific regions of an image by substituting the cross-attention maps (CAMs) corresponding to the target edit words within the source prompt. Furthermore, Instruct-Pix2Pix [1] refines an instruction-based model by creating an image translation training dataset using P2P-generated images.

Despite their advancements, current TIE algorithms are not without limitations. As illustrated in Figure 1, the prevalent TIE methods encounter challenges when editing multiple objects within an image. The complexity of multi-object editing amplifies issues such as object loss (e.g., a missing apple), absence of object attributes (like spots), and incomplete representations of the background. Building on the analyses of earlier works [4, 5, 15, 18, 34], these deficiencies

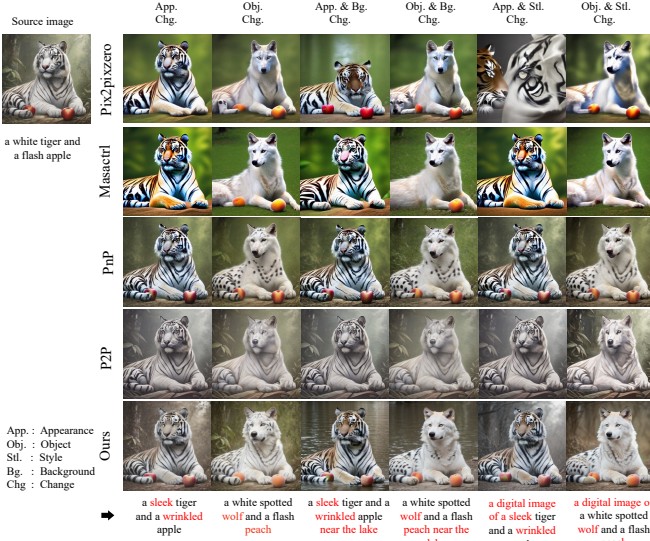

**Figure 1: Examples of TIE results. Compared to previous methods such as Pix2pixzero [23], Masactrl [2], PnP [33], P2P [10], our method successfully corrects the editing errors of object loss, attribute absence, and incomplete backgrounds while preserving the spatial structure of the original image.**

in editing accuracy are predominantly due to the imprecision of CAMs in representing multiple objects.

In this paper, we introduce VICTORIA[2], a novel approach that enhances training-free TIE by leveraging linguistic knowledge to address the issue of lost editing targets, such as objects, attributes, and backgrounds, especially in multi-target image editing scenarios. VICTORIA improves upon diffusion-based TIE by manipulating intermediate representations within attention layers to generate images, while taking linguistic insights into account. Rather than replacing CAMs, our method utilizes keys and queries in self-attention layers of the source image. This approach maintains spatial consistency between the source image and the edited target image. While this technique successfully achieves spatial consistency, addressing the inaccuracy of CAMs during image generation remains a challenge. To tackle this, VICTORIA carries out a syntactic analysis of the target editing text to discern word relationships. Utilizing these syntactic relations, we can guide the refinement of CAMs in the diffusion model, thereby improving the accuracy of edits across multiple targets without modifying the underlying TIS model. In the experiments, VICTORIA demonstrates superior performance, aligning edits more precisely on both new and established public benchmark datasets.

In summary, the contributions of our paper are as follows:

- We have developed a novel TIE algorithm that incorporates linguistics-based tracking. This algorithm is designed to be training-free, offering a plug-and-play solution that is compatible with current popular TIE methodologies.

- For TIE tasks involving multiple objects, we leverage linguistic analysis to enhance the editing of individual objects, thereby improving the capability for multi-object editing. We also fill the gap in TIE by crafting a dataset specifically for multi-object editing tasks.
- VICTORIA achieves the state-of-the-art (SOTA) performance across seven TIE benchmark datasets, underscoring the pivotal role and potency of linguistic analysis in enhancing the capabilities of TIE.

## 2 Related Works

### 2.1 Diffusion-based TIS Models

In recent years, significant advancements have been achieved in text-driven image processing with Generative Adversarial Networks (GANs) [9, 14, 24] and vision-language models such as CLIP model [25]. While GANs have exhibited proficiency in handling in-domain data, they struggle with larger and more diverse datasets. Diffusion-based models like DALL-E 2 [26], Imagen [30], and Stable Diffusion [28] have been instrumental in propelling TIS forward by demonstrating exceptional prowess in synthesizing high-fidelity images. Despite their success, these models do not inherently possess TIE capabilities for the images they generate. Our work explores training-free TIE algorithms that build upon diffusion-based TIS models, thereby extending their editing functionality.

### 2.2 Text-guided Image Editing

TIE is a crucial task involving modifying an input image with requirements expressed by texts. In the literature, TIE approaches can be summarized into two categories: training-free and training-based. Training-free methods [2, 5, 10, 17, 18, 22, 23, 33] are designed to manipulate image generation during the denoising process. For instance, SDEdit [17] innovatively adds noise to a selected guide image to serve as the initial noise, yielding notable results. P2P [10] alters cross-attention maps to control the relationship between an image's spatial layout. Training-based methods craft new, ideal images by tweaking the model with domain-specific insights [7, 12, 15, 29] or by integrating supplementary guidance data [1, 19, 35]. Specifically, ControlNet [35] and T2I-Adapter [19] allow users to navigate the direction of image generation using input images by altering additional network modules. However, current TIE methods, either training-free or training-based, need more analysis of the relationships in CAMs. Our paper investigates the roles of entities and their modifiers of prompts in TIE through syntactic analysis.

## 3 VICTORIA: Proposed Approach

Given an image, which can be a synthesis image generated from prompts or a real image, our objective is to guide the diffusion model to perform multi-object, multi-attribute editing based on the prompt. Our goal is to ensure that the edited image preserves the spatial structure of the source image while conforming to the target text description. In this task, maintaining spatial consistency between images is fundamental. Inspired by prior work [2, 10, 16, 33], we accomplish this through control over the self-attention mechanism. Additionally, two other aspects require careful attention

---

[2]VICTORIA stands for attenti**V**e l**I**nguisti**C** **T**racking f**O**r t**R**aining-free text-gu**I**ded im**A**ge editing. Source code and datasets will be made available upon paper acceptance.

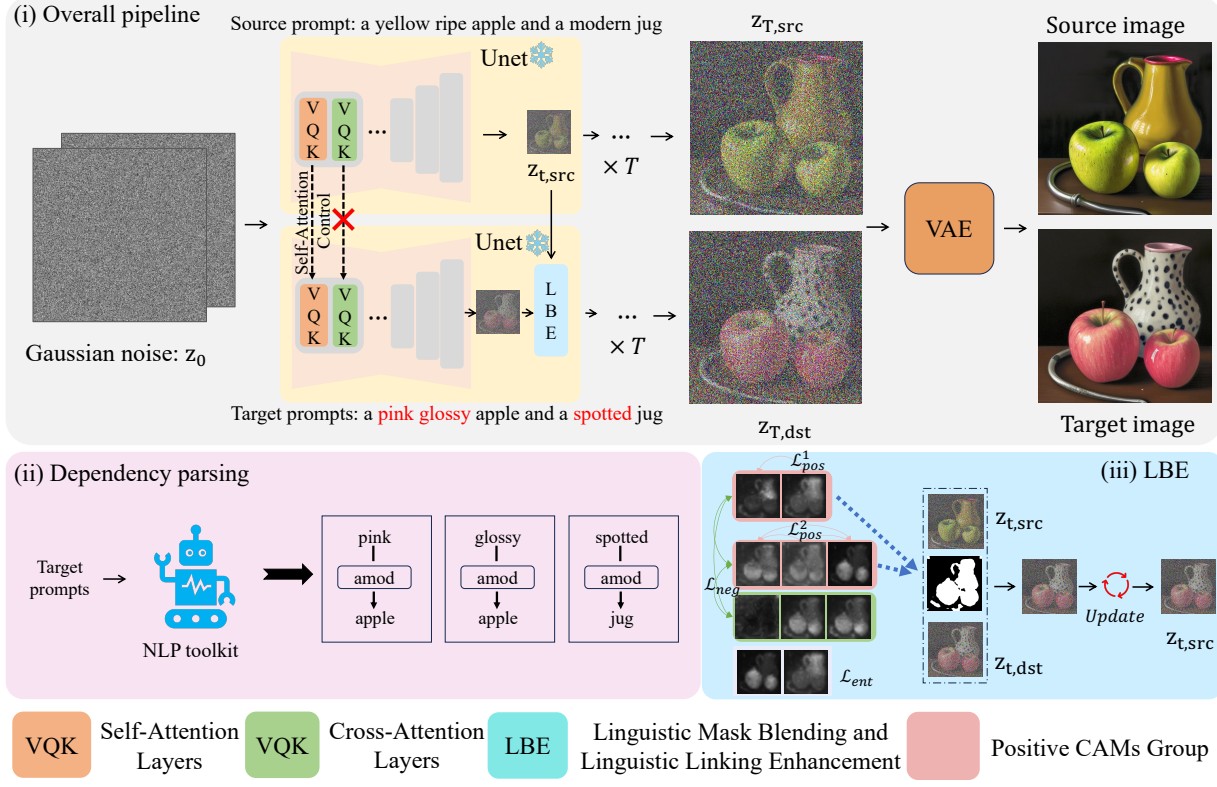

**Figure 2: An overview of the VICTORIA framework: VICTORIA includes three key components: input text prompt processing, output region restriction, and enhancement. We use dependency parsing to extract modifier-head relations from the input text. Self-Attention layers preserve the structural integrity of source images. LBE protects unchanged regions and enriches generative outcomes in focal areas.**

to guarantee accurate image editing. First, unlike preceding methods [2, 10, 16, 33] that depend on the diffusion model's inherent cross-attention layer, our approach actively intervenes in the CAM generation process during the creation of the target edited image. This ensures a precise alignment between input text and target image at the cross-attention level. Second, it is crucial to preserve the regions of the image that do not require editing, which is achieved through modified masks informed by the cross-attention.

An overview of VICTORIA is depicted in Figure 2, which illustrates how VICTORIA is stratified into three distinct parts: input processing, output region restriction, and enhancement. Here, the "Self-Attention Control" module is dedicated to preserving the structural integrity of source images. The "Linguistic Linking Enhancement" part enriches the generative outcomes in the focal editing areas, making it particularly adept for multi-object editing scenarios. Lastly, the "Linguistic Mask Blending" component is designed to safeguard the information in regions that remain unaltered.

### 3.1 Self-Attention Control

A primary challenge in TIE is preserving the spatial layout between the source and target images. Contemporary editing methods, such as P2P [10], typically involve the replacement of CAMs during the generation process. However, as identified in previous research [16],

replacing CAMs may inadvertently transfer an excess of information from the source image, potentially leading to undesirable outcomes. Moreover, P2P relies on textual consistency between source and target prompts, thus complicating its direct use for editing images that lack text prompts.

Figure 3 illustrates the impact of replacing CAMs and SAMs on TIE, where a car edited using CAMs exhibits loss of structure and color fidelity. While editing with both CAMs and SAMs improves structural reconstruction, it falls short in the accurate translation of color. Conversely, selectively replacing SAMs at a low ratio produces images that align more closely with the target prompt. Additionally, Figure 4 presents the editing outcomes across different self-attention layers. The findings show that replacements across all layers result in a target image that mirrors the source. In contrast, limiting modifications to the middle layers significantly alters the spatial structure of the target image.

Based on the above observations, we extract queries ($Q_{src}$) and keys ($K_{src}$) from self-attention layers, setting us apart from previous image editing methods [2, 10, 16, 33]. The process is formulated as follows:

$$M_{dst} = Softmax(\frac{Q_{src} \cdot K_{src}{}^{T}}{\sqrt{d}}) \qquad (1)$$

where $M_{dst}$ is the SAM of the target image during the editing process, $d$ is the dimension of keys and queries. The final output is conceptualized as a composite feature, which synergistically blends the structural characteristics of the source image with the feature attributes of the target image, denoted as: $f = M_{dst} \cdot V_{dst}$ where $V_{dst}$ refers to the self-attention values of the target image. These elements are seamlessly integrated into the corresponding self-attention layers for generating the target image. Compared with directly substituting SAMs of fixed layers, the advantage of our method is that it can accommodate popular acceleration algorithms such as flash-attention [6], which do not explicitly compute attention maps.

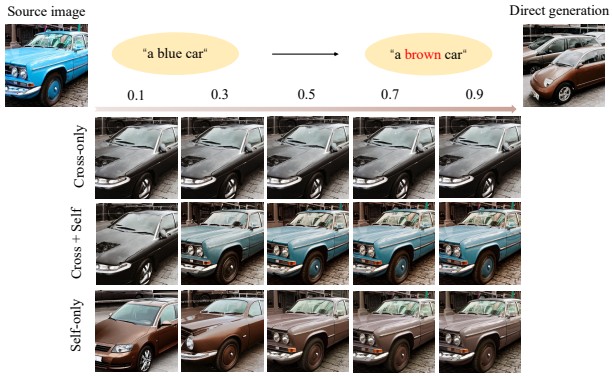

**Figure 3: Results on three types of attention map replacements at different replace step ratios.A higher ratio denotes a greater number of replacement steps.**

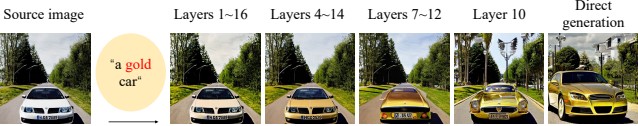

**Figure 4: Results of self-attention replacements in different layers of the diffusion model.**

## 3.2 Linguistic Linking Enhancement

After ensuring the spatial consistency between source and target images, it is important to also ensure the consistency between the target image and the text description (i.e., prompt). However, in contrast to previous methods ([2, 10, 16, 33]), which solely entrusted the task of integrating the text to the diffusion model's built-in cross-attention layer, the inaccuracies introduced by the diffusion model's cross-attention layer may lead to the failure of integrating text and image, especially in the case of multi-object, multi-attribute editing (as shown in the introduction). To address these challenges, we introduce the Linguistic Linking Enhancement (LLE) technique aimed at improving TIE in such demanding scenarios by refining the editing process. The core concept involves a subtle optimization of the target CAMs during the editing process, rather than adjusting the parameters of the TIS model itself. This step is highly efficient

and is executed during model inference, thereby maintaining our approach as training-free.

We first extract modifier-head relations from the input prompt by dependency parsing [11]. We define $S$ as the collection of modifier-head pairs from the parsing result. The LLE technique is applied to all CAMs with respect to $S$. Furthermore, we denote by $d(M_m, M_n)$ a distance measure between the CAMs $M_m$ and $M_n$ corresponding to the modifier-head pair $(s_m, s_n) \in S$, where $s_m$ and $s_n$ represent the modifier and the headword, respectively. For instance, considering the prompt "a pink glossy apple and a spotted jug", the set $S$ contains three pairs: 1) $s_m$ = "pink" and $s_n$ = "apple", 2) $s_m$ = "glossy" and $s_n$ = "apple", and 3) $s_m$ = "spotted" and $s_n$ = "jug".

In LLE, we initially construct a positive loss to minimize the CAM distance between modifiers and headwords, defined as:

$$\mathcal{L}_{pos}(M, S) = \sum_{(s_m, s_n) \in S} d(M_m, M_n) \tag{2}$$

For instance, this mechanism encourages the CAMs with respect to "pink" and "apple" to be brought closer. Additionally, we design a loss function that quantifies the disparity between the words in $S$ and the rest of the words present in the target prompt, promoting the dissociation of word pairs that are not syntactically connected. Let $U$ represent the collection of words in the target prompt not included in $S$, and $M_u$ denotes the CAM corresponding to an unrelated word $u \in U$ (e.g., "a", "and"). The negative loss is hence defined as follows:

$$\mathcal{L}_{neg} = - \sum_{(s_m, s_n) \in \tilde{S}} \sum_{u \in U} d(M_m, M_u) + d(M_u, M_n) \tag{3}$$

In this context, we employ the symmetric Kullback–Leibler (KL) divergence as the distance measure to evaluate the proximity between two normalized CAMs.

Furthermore, to encourage the CAMs of entity words (e.g., "apple", "jug") in modifier-head pairs to have a high activation value, thereby focusing the attention map more precisely on the corresponding object region, we define the attention loss as follows:

$$\mathcal{L}_{ent} = \max_{n \in N} \mathcal{L}_n \quad \text{where} \quad \mathcal{L}_n = 1 - \max(M_n). \tag{4}$$

The overall loss function for our LLE task is then given by:

$$\mathcal{L}_{LLE} = \mathcal{L}_{pos} + \mathcal{L}_{neg} + \mathcal{L}_{ent}. \tag{5}$$

Our optimization strategy draws parallels with the approach in [27], where the loss is predicated on pairs of all linguistically-related words for the task of image generation. In contrast, LLE focuses specifically on edited words and their linguistically-associated counterparts within the target prompt. It also aims to enhance the specificity of CAMs related to objects, particularly tailored for TIE.

## 3.3 Linguistic Mask Blending

The final aspect of our framework addresses the preservation of non-editing areas. To combat contamination in non-target regions during the denoising process, we have adopted a strategy that exerts control over the background using masks, inspired by DAAM [32]. Our method dynamically generates editing masks by capitalizing on CAMs during the generation process of the target image.

We denote $S_w$ as the set including the editing word $w$, as well as its associated modifiers and headword. Taking the prompt "a photo

of a colorful ruffed grouse" as an example, $S_w$ with respect to the editing word "colorful" encompasses not only "colorful" itself but also "ruffed" and "grouse". We let $W$ represent the set of all editing words and define $S_W = \bigcup_{w \in W} S_w$ to represent the aggregation of words that are semantically tied to any word in $W$. Moreover, we represent each CAM associated with a term $s_i \in S_W$ as $M_i$. The linguistic blending mask is generated as follows:

$$Mask = \tau \cdot \left( \bigcup_{i=1}^{|S_W|} M_i \right) \quad (6)$$

where $\tau$ is a threshold empirically chosen from the set $\{0.3, 0.4, 0.5\}$.

Let $z_{src}$ and $z_{dst}$ be the latent codes of the source and target images, respectively, during the diffusion denoising process. The resultant latent codes for the generated image (denoted as $z_{dst}^*$) can be determined by the following equation:

$$z_{dst}^* = (1 - Mask) \cdot z_{src} + Mask \cdot z_{dst} \quad (7)$$

This equation facilitates a balanced blending of the source and target images, taking into account linguistic knowledge. For a visual representation of this blending process, readers are directed to Figure 5 for a simple example.

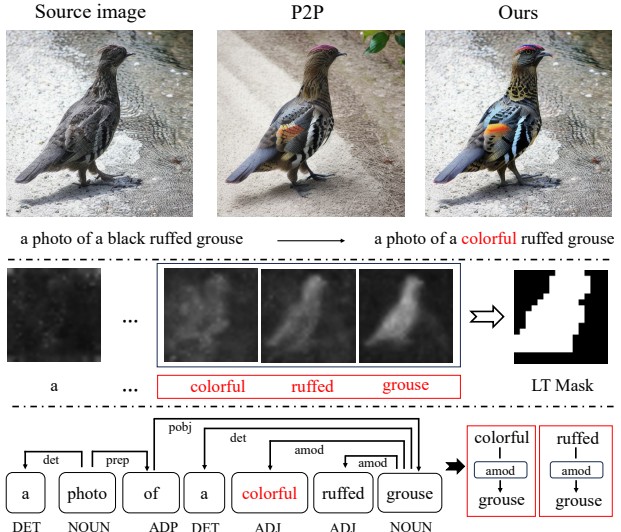

a photo of a black ruffed grouse ⟶ a photo of a colorful ruffed grouse

**Figure 5: An example of Linguistic Mask Blending.**

## 3.4 Summary of VICTORIA

We summarize the VICTORIA algorithm in the form of pseudo-codes. Let $DM(z_t, P, t)$ be the computation of step $t$ of the diffusion process, which outputs the latent code $z_{t-1}$, $Q_{src}, K_{src}$ (from self-attention layers) and the CAM $M_{cross}$ (from cross-attention layers, omitted if not used). We denote by $DM(*)$ the diffusion step where we override $Q_{dst}, K_{dst}$ with additional given $Q_{src}, K_{src}$, but keep the values $V$ from the editing image. Formally, Algorithm 1 is tailored for editing synthetic images. Algorithm 2 is adept at managing scenarios encompassing real-world images without source prompts, skillfully transmuting source images into latent noise

representations via DDIM-inversion [31]. This conversion process through DDIM-inversion effectively transforms real images into latent noises, which then act as initial latent codes for source images.

---

**Algorithm 1** VICTORIA (Synthetic Image)

**Input:** $P_{src}$: source prompt; $P_{dst}$: target prompt; $W$: editing words; $T$: number of steps;
**Output:** $I_{src}$: source image; $I_{dst}$: edited image;
1: $z_{T,src} \sim \mathcal{N}(0,1)$; # Gaussian noise
2: $z_{T,dst} \leftarrow z_{T,src}$;
3: $S \leftarrow DependencyParsing(P_{dst}, W)$;
4: **for** $t = T, T-1, ..., 1$ **do**
5: $\quad z_{t,dst} \leftarrow$ update $z_{t,dst}$ by $L_{LLE}$;
6: $\quad z_{t-1,src}, Q_{src}, K_{src} \leftarrow DM(z_{t,src}, P_{src}, t)$;
7: $\quad z_{t-1,dst}, M \leftarrow DM(z_{t,dst}, P_{dst}, t)\{Q_{dst}, K_{dst} = Q_{src}, K_{src}\}$;
8: $\quad$ **if** $Blending$ **then**
9: $\quad\quad Mask \leftarrow Eq.(6)(S, M)$;
10: $\quad\quad z_{t-1,dst}^* \leftarrow$ update $z_{t-1,dst}$ by $Eq.(7)$;
11: $\quad$ **end if**
12: **end for**
13: $(I_{src}, I_{dst}) \leftarrow VAE(z_{0,src}, z_{0,dst}^*)$;
14: **Return** $(I_{src}, I_{dst})$.

---

**Algorithm 2** VICTORIA (Real Image)

**Input:** $P_{dst}$: a target prompt; $I_{src}$: real image; $W$: editing words;
**Output:** $I_{res}$: reconstructed image; $I_{dst}$: edited image;
1: $\{z_{t,src}\}_{t=0}^T \leftarrow DDIM - inv(I_{src})$;
2: $z_{T,dst} \leftarrow z_{T,src}$;
3: $S \leftarrow DependencyParsing(P_{dst}, W)$;
4: **for** $t = T, T-1, ..., 1$ **do**
5: $\quad z_{t,dst} \leftarrow$ update $z_{t,dst}$ by $L_{LLE}$;
6: $\quad z_{t-1,src}, Q_{src}, K_{src} \leftarrow DM(z_{t,src}, t)$;
7: $\quad z_{t-1,dst}, M \leftarrow DM(z_{t,dst}, P_{dst}, t)\{Q_{dst}, K_{dst} = Q_{src}, K_{src}\}$;
8: $\quad$ **if** $Blending$ **then**
9: $\quad\quad Mask \leftarrow Eq.(6)(S, M)$;
10: $\quad\quad z_{t-1,dst}^* \leftarrow$ update $z_{t-1,dst}$ by $Eq.(7)$;
11: $\quad$ **end if**
12: **end for**
13: $(I_{res}, I_{dst}) \leftarrow VAE(z_{0,src}, z_{0,dst}^*)$;
14: **Return** $(I_{res}, I_{dst})$.

---

## 4 Experiments

### 4.1 Datasets and Experimental Settings

Considering the scarcity of public datasets for verifying the effectiveness of Text-Image Editing (TIE) algorithms in scenarios involving editing of multiple objects, we developed two types of evaluation datasets for assessing our approach in both synthetic and real-image scenarios, which will be released in public. The first, a synthetic image dataset named DVMP-edit-fake[3], was specifically crafted to test the handling of editing adjectives and objects. It consists of 200 prompt pairs. For real-image scenarios, we introduced the DVMP-edit-real dataset, which comprises 808 image-prompt pairs. Moreover, for additional validation, we leveraged

---

[3]The DVMP-edit-fake dataset is a variant of the Diverse Visual Modifier Prompts (DVMP) dataset [27].

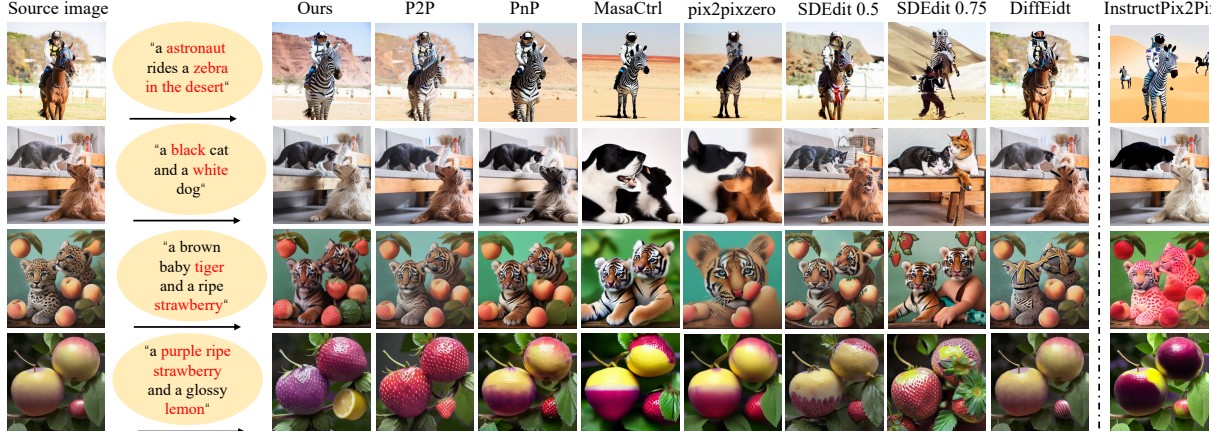

**Figure 6: Comparison to prior works. Left to right: source image, target prompt, our result, P2P [10], PnP [33], MasaCtrl [2], SDEdit [17] w/ two noising levels, DiffEdit [5], pix2pixzero [23] and InstructPix2Pix [1]. Only InstructPix2Pix is the training-based method.**

several publicly available image editing datasets created by in prior research [13, 16, 33], including PIE-Benchmark [13], Car-color-real [16], Car-color-fake [16], ImageNet-R-TI2I [33], and Wild-TI2I-real [33]. Human evaluation, CLIP Score (CS), and Directional CLIP Similarity (CDS) [8, 25] are utilized to quantitatively analyze and compare our method against prevalent text-guided image editing (TIE) algorithms. The underlying TIS model employed in our experiments is the Stable Diffusion 1.5[4]. We generated experimental results for baselines using publicly available code from their original papers, ensuring consistency with unified random seeds.

## 4.2 Image Editing Results

Figure 7 demonstrates the remarkable editing capabilities of our method across a range of scenarios. Our technique skillfully handles single attribute modifications, adeptly manages changes to attributes and backgrounds within generated scenes, and accurately replaces target entities. It also enables subtle shifts in visual style. Crucially, when confronted with complex editing tasks that involve multiple entities, our approach seamlessly executes simultaneous transformations on numerous entities and their attributes. Moreover, our method's independence from prompt alignment offers users the flexibility to precisely refine target images. This can be achieved through advanced prompt engineering techniques, including the utilization of prompt optimization tools [3, 20].

## 4.3 General Comparison

In this section, we first compare our method with SOTA methods [1, 2, 5, 10, 17, 23, 33] over DVMP-edit-real, DVMP-edit-fake datasets and PIE-Benchmark. Experimental results are presented in Figure 6 and Table 1. As shown in Figure 6, our method successfully converts real and synthetic images to target ones. In all examples, our method achieves high-fidelity editing that aligns with target prompts while preserving the structural information of source images to the maximum extent possible. Quantitative results

| Methods | DVMP-edit-real | | DVMP-edit-fake | | PIE-Benchmark | |
|---|---|---|---|---|---|---|
| | CS ↑ | CDS ↑ | CS ↑ | CDS ↑ | CS ↑ | CDS ↑ |
| DiffEdit | 26.92 | 0.0999 | - | - | - | - |
| SDEdit | 28.54 | 0.1387 | - | - | 28.52 | 0.0946 |
| Pix2pixzero | 26.75 | 0.2073 | 25.27 | 0.0991 | 28.56 | 0.1068 |
| P2P | 27.88 | 0.1479 | 25.56 | 0.2363 | 27.87 | 0.1440 |
| PnP | 28.05 | 0.1698 | 25.29 | 0.2605 | 28.21 | 0.1226 |
| Masactrl | **29.26** | 0.2034 | 23.59 | 0.0755 | **29.15** | 0.1452 |
| Ours | 28.95 | **0.2087** | 26.64 | **0.2784** | 28.59 | **0.1703** |

**Table 1: Quantitative experimental results over DVMP-edit-real, DVMP-edit-fake datasets and PIE-Benchmark. CS: Clip score [25] and CDS: Clip Directional Similarity [8, 25].**

are presented in Table 1. From Table 1, our method outperforms all others in terms of CDS, indicating that our method excels in preserving the spatial structure and performing editing according to the requirements of target prompts, yielding superior results.

We further carry out human evaluations over the Wild-TI2I-real and ImageNet-R-TI2I datasets. In detail, from a pool of 228 examples, we invite a group of participants to choose what they consider the best TIE results by assessing source images alongside the target prompts provided. The findings are depicted in Figure 8, illustrating the percentage of images the participants favor most. The figure showcases a comparison of user preferences for Ours and two strong baselines (i.e., P2P and MasaCtrl), as well as instances where the performance of the three methods appears to be on par. The result shows the superiority of our approach, with the highest rate at 43.64%.

## 4.4 Ablation Studies

**Linguistic Linking Enhancement.** On DVMP-edit-real, we conducted ablation studies by incorporating different optimization losses proposed in Linguistic Linking Enhancement (LLE) under self-attention control , with results detailed in Table 2 and Figure 9. Table 2 reveals that different optimization losses can improve the text-image consistency of edits while preserving spatial coherence, but with $L_{LLE}$ performs the best. Figure 9(a) demonstrates that when only using self-attention control (Self-Ctrl) for image editing,

---
[4]https://huggingface.co/runwayml/stable-diffusion-v1-5

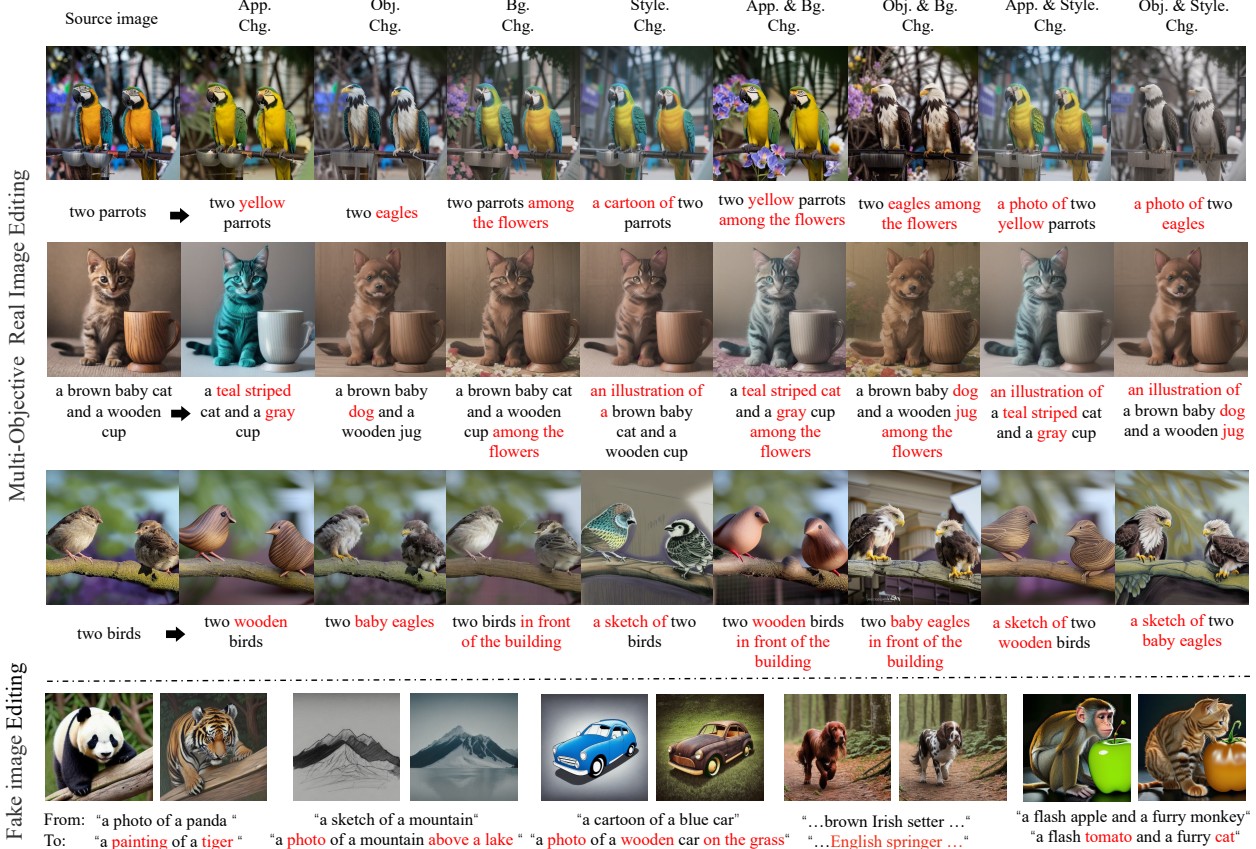

Figure 7: Image editing results of VICTORIA using the current popular Text-to-Image Synthesis Model.

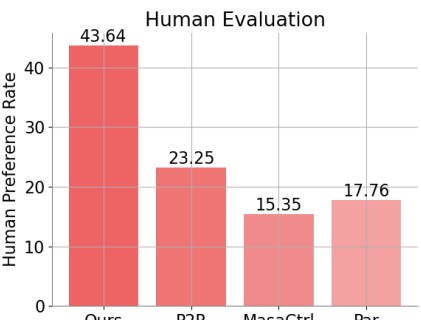

Figure 8: Human evaluation results over Wild-TI2I-real and ImageNet-R-TI2I.

both $L_{ent}$ and $\{L_{pos}, L_{neg}\}$ optimization result in missing object attributes or semantic misalignment. However, when using $L_{LLE}$ loss, these errors are corrected. For example, in Figure 9(a), $L_{LLE}$ corrected the attributes of the "teal Striped cat"during the target editing. For single object edits, LLE enhanced object information (Figure 9(c)) and corrected attribute alignment errors (Figure 9(d)). For instance, in Figure 9(d), both Self-Ctrl and Self-Ctrl + $L_{ent}$ wrongly aligned the attribute "yellow" to the "blanket," which our method corrected.

We further demonstrate the example of using LLE to correct the alignment of CAMs in Figure 10. As demonstrated in Figure 10, the leading cause of these failures is the text-image misalignment occurring in cross-attention layers. For instance, the word "strawberry" is incorrectly associated with the "candle" in the source image. The regions corresponding to the words "checkered" and "candle" are erroneously mapped onto "strawberry". Our method addresses these inaccuracies by employing LLE on CAMs to realign textual descriptions with appropriate image regions. This adjustment effectively corrects the editing errors, creating an edited image that accurately reflects textual descriptions.

| Methods | CS ↑ | CDS ↑ |
|---|---|---|
| Self-Ctrl | 28.40 | 0.1604 |
| Self-Ctrl + $L_{ent}$ | 28.48 | 0.1608 |
| Self-Ctrl + $L_{pos}$ + $L_{neg}$ | 28.69 | 0.1807 |
| Self-Ctrl + $L_{pos}$ + $L_{neg}$ + $L_{ent}$ | **28.95** | **0.2087** |

Table 2: Ablation experiments in multi-object editing over DVMP-edit-real. Self-Ctrl is the method in Section 3.1.

**Linguistic Mask Blending.** We execute ablation studies on Car-color-fake and Car-color-real, with findings in Tables 3. During these studies, we benchmark the performance of P2P, which involves edits on CAMs against our approach, which operates on

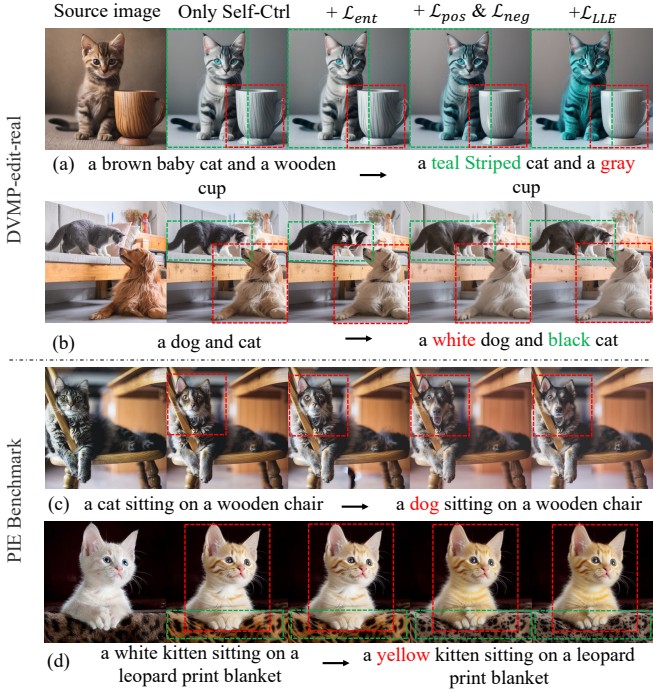

Figure 9: Ablation results in multi-object editing with different optimization losses. Self-Ctrl is the method in Section 3.1

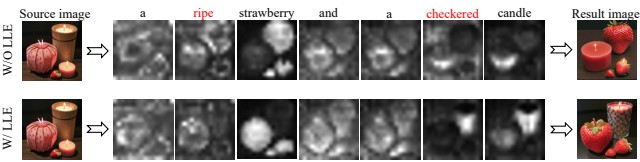

**Figure 10: Example on how Linguistic Linking Enhancement corrects and enhances CAMs in multi-object editing.**

Self-Attention Control (Section 3.1). Additionally, we evaluate various blending strategies: no blending, blending focused solely on editing words (only editing words) and blending encompassing both editing words and their dependent counterparts (our approach). Our method outperformed P2P in terms of effectiveness, highlighting the advantages of our blending strategy, which proves beneficial for both P2P and our method. Conclusion also can draw from Figure 11. As shown in Figure 11, direct modifications to elements within self-attention layers can yield images that align with editing instructions but can alter contents in areas not intended for editing. To address this issue, we construct linguistic masks to preserve contents in non-editing areas. We successfully preserve source images' non-edited areas by employing this mask blending strategy.

## 5 Limitation

The capabilities of the underlying TIS model may bind VICTORIA's performance. While VICTORIA can enhance the consistency of edited images, it encounters some limitations as the complexity of

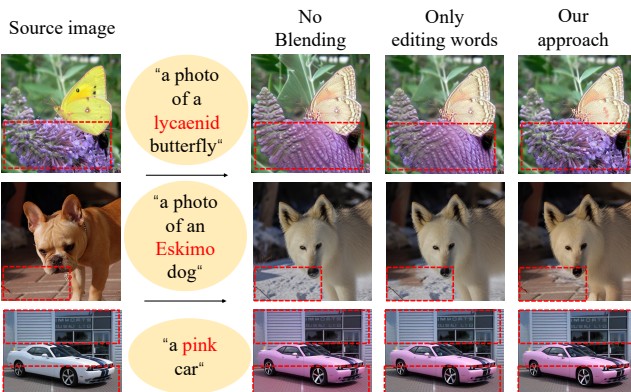

**Figure 11: Analysis on blending within editing. From left to right, source image, editing result of no blending, blending only on editing words and our approach.**

| Method | Car-color-fake | | Car-color-real | |
|---|---|---|---|---|
| | CS ↑ | CDS ↑ | CS ↑ | CDS ↑ |
| P2P | 25.33 | 0.2577 | 24.65 | 0.2593 |
| P2P + only editing words | 25.34 | 0.2634 | 24.66 | 0.2683 |
| P2P + our mask blending | 25.32 | 0.2664 | **24.68** | 0.2726 |
| Self-Ctrl | _25.37_ | 0.2682 | 24.66 | 0.2805 |
| Self-Ctrl + only editing words | 25.36 | _0.2760_ | 24.65 | _0.2913_ |
| Our approach | **25.39** | **0.2765** | _24.67_ | **0.2925** |

**Table 3: Ablation experiments of linguistic mask blending. Self-Ctrl is the method in Section 3.1.**

entities and the number of modifier words increase. The generation process sometimes fails when the underlying TIS model is unable to generate images aligned with the target prompt. VICTORIA is not good at removing an object by removing the corresponding word in the prompt. In actual image editing scenarios, the latent code derived from DDIM-inversion [31] carries priors of the source image, which can be unfavorable for regenerating images to fit certain prompts. Addressing these issues will be our future work.

## 6 Conclusion

In conclusion, our proposed VICTORIA approach significantly advances the field of TIE. By leveraging linguistic insights, VICTORIA effectively addresses common challenges in existing TIE methods, especially in the case of multi-object, multi-attribute editing, such as maintaining spatial consistency and aligning attention maps with textual semantics. Our innovative linguistic mask blending technique and tailored loss function have demonstrated their effectiveness, exhibiting superior performance across multiple datasets. This work emphasizes the critical role of linguistic analysis in multi-object entity image editing and paves the way for new avenues of research and development.

## Acknowledgments

This work is partially supported by Alibaba Cloud through the Research Talent Program with South China University of Technology, and the Guangdong Provincial Key Laboratory of Human Digital Twin under Grant 2022B1212010004.

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
