# OpenReview forum: "Attentive Linguistic Tracking in Diffusion Models for Training-free Text-guided Image Editing"
_acmmm.org/ACMMM/2024/Conference — MM2024 Poster_

### Official Review · Reviewer_D554 · 2024-05-12

**Rating:** 5
**Confidence:** 4

**Summary:**

A novel approach is proposed in the paper to handle with TIE problem via self-attention control. In the framework proposed in the paper, multiple words amended in the editing prompt can correspond to different objects in the input image. More specifically, *Linguistic Linking Enhancement* is proposed to optimize the target CAMs instead of re-training or fine-tuning the diffusion models.

**Strengths:**

1. Changes in multiple words in the prompt can be correspondingly reflected to the corresponding targets, breaking through the lack of relational understanding in LDMs under the control of cross-attention in the past and spatial information is simultaneously preserved.

2. Sufficient visual results and extensive experiments demonstrate the ability of the approach.

3. *Linguistic Mask Blending* enhances the ability of the framework to preserve the irrelevant regions, which is also remarkable. In addition, the editing results show that the abstract adjectives can be visualized in some ways, which has not been seen in previous works to the best of my knowledge.

**Limitations:**

1. The author may show some of the failure cases in the paper. It can make the specific scenarios the model applies clearer to the reader, thus making it more attractive and objective.

2. The NLP toolkit in this framework can be replaced with LLM and the approach can be evaluated by MLLM for accuracy when editing for multiple objects, which can enhance the innovation and improve the persuasiveness of the paper.

However, I have the following questions about the ability of the approach:

1. I am curious to know if *VICTORIA* can perform like approaches in papers like "Imagic: Text-Based Real Image Editing with Diffusion Models"[CVPR'2024] or "Edit One for All: Interactive Batch Image Editing"[CVPR'24]?

2. If I would like to remove an object from a picture, can I do it by removing the corresponding word in the prompt? (or replace the word with some special tokens?)

**Suitability:**

3

---

### Official Review · Reviewer_wbPc · 2024-05-21

**Rating:** 4
**Confidence:** 4

**Summary:**

The paper introduces VICTORIA, a novel training-free text-guided image editing approach that incorporates linguistic knowledge to enhance the editing precision of multiple target objects within images by manipulating attention maps during the generation process.

**Strengths:**

VICTORIA addresses the challenge of multi-object editing by refining cross-attention maps through syntactic analysis of the target editing text, which is an innovative way to improve the alignment of textual descriptions with image content.

**Limitations:**

W1: As shown in Table 3, the improvement brought about by mask blending seems to be quite limited. However, it is the main contribution of this paper, questioning whether the enhancement all stems from Linguistic Linking Enhancement.

W2: The author lacks discussion on inference efficiency and how the computational cost of the method presented in this paper differs from previous methods.

W3: Formatting issue. The expression in this article has some inadequacies, as well as some issues with the formatting of symbols, for instance, the subscript of Loss should be written in upright type.

**Suitability:**

3

---

### Official Review · Reviewer_3sWM · 2024-05-24

**Rating:** 3
**Confidence:** 4

**Summary:**

The paper focusses on the task of text-guided image editing, where the authors have proposed a training-free method for editing images by utilising queries and keys from the self attention maps and also optimises the maps (guided by language parsing performed on prompt to capture relations)

**Strengths:**

well written and easy to follow paper; training-free approach; covers multiple scenarios of image editing like appearance, background, multi-object editing and so on

**Limitations:**

1. It is unclear how the structural consistency part obtained via utilizing the keys and queries is different from the similar insight proposed in this paper on cross-image attention for appearance transfer [1]
2. Prompt2prompt's limitation on the requirement of knowing the prompt corresponding to the generated image is valid. But the authors should also try how their results compare will Null Text Inversion [2]
3. It is unclear why would the multi-object editing be a challenge with existing baselines?
4. Some results in the main paper (e.g., the "two birds" case in Figure 7 where the edit asks for "in front of the building) are not good/convincing
5. The proposed approach seems to be a combination of multiple existing baselines like [1] and [3] where [3] as well utilises clues from language parsing to optimize the attention maps. I agree that the authors maybe the first ones to show the applicability of such a method in the context of image editing, but more clarity into their novel technical contributions is required.

[1] Cross-Image Attention for Zero-Shot Appearance Transfer
[2] Null-text inversion for editing real images using guided diffusion models
[3] Linguistic Binding in Diffusion Models: Enhancing Attribute Correspondence through Attention Map Alignment

**Suitability:**

2

---

### Official Review · Reviewer_RQVf · 2024-05-24

**Rating:** 3
**Confidence:** 3

**Summary:**

The paper presents a text-to-image editing approach (VICTORIA) that leverages linguistic mask blending technique to achieve multi-object, multi-attribute editing tasks, and better preserve irrelevant content compared to prompt2prompt (P2P). VICTORIA consists three major modules: self-attention module, linguistic linking enhancement, and linguistic mask blending. Experiments are performed on various editing sets created by prior work[13, 16, 33].

**Strengths:**

- VICTORIA can preserve the irrelevant content of the image better compared to P2P.

- Following P2P, VICTORIA operates in a training-free manner, which could be helpful in real-world applications.

**Limitations:**

- Although the paper mentions that a significant advantage of VICTORIA over previous methods is the preservation of text-irrelevant content, the proposed linguistic masking mechanism seems not that satisfying. For example, in columns 1, 2, 5, and 6 of Figure 1, the background parts should be preserved, yet the generated images still have different backgrounds. Considering the advancements in instance segmentation, many researchers currently use Grounded-SAM to preserve text-irrelevant regions. It is unclear why this paper did not adopt this approach or compare it.

- Another contribution of the paper is the ability of multi-object editing. On one hand, multi-object editing can be easily achieved by editing objects one-by-one. On the other hand, existing methods (*e.g.*, P2P, PnP) seem also have multi-object editing ability. In addition, when an image contains multiple objects, VICTORIA seems to randomly select an object to edit, and it is unclear how to precisely edit the target object (*e.g.*, Figure 6, "a purple ripe strawberry.").

- Table 1: Standard evaluation metrics such as DINO, FID, and GPT-4V are not reported.

- There is no discussion of the method's limitations. For example, VICTORIA does not seem to support non-rigid editing, like turning a sitting dog into a running dog as Imagic, Dreambooth did. The preservation of text-irrelevant content is also not very accurate.

**Suitability:**

2

---

### Meta-Review · Area_Chair_JYE9 · 2024-07-03

**Recommendation:** Accept (Poster)
**Confidence:** 5

**Metareview:**

The final ratings of this paper are two borderline rejects and two weak accept. The main concerns from the reviewers are novelty and experimental results. The rebuttal addressed some questions, the AC agrees with the reviewers that the paper has shown sufficient visual results and extensive experiments demonstrate the approach's ability. Therefore, the AC recommends accepting this paper.

---

### Meta-Review · Senior_Area_Chairs · 2024-07-10

**Recommendation:** Accept (Poster)
**Confidence:** 4

**Metareview:**

This paper received mixed ratings initially. After rebuttal, 2 reviewers tend to accept the paper, one did not sumbit the final rating, one still questioned the novelty of the paper. SAC and AC carefully read the paper, reviews and rebuttal and recommend acceptance of the paper.